# Decoding Brain Signals in a Neuromorphic Framework for a Personalized Adaptive Control of Human Prosthetics

**DOI:** 10.3390/biomimetics10030183

**Published:** 2025-03-14

**Authors:** Georgi Rusev, Svetlozar Yordanov, Simona Nedelcheva, Alexander Banderov, Fabien Sauter-Starace, Petia Koprinkova-Hristova, Nikola Kasabov

**Affiliations:** 1Institute of Information and Communication Technologies, Bulgarian Academy of Sciences, 1113 Sofia, Bulgaria; georgi.rusev@iict.bas.bg (G.R.); svetlozar.yordanov@iict.bas.bg (S.Y.); simona.nedelcheva@iict.bas.bg (S.N.); aleksandar.banderov@iict.bas.bg (A.B.); 2Univ. Grenoble Alpes, CEA, Leti, F-38000 Grenoble, France; fabien.sauter@cea.fr

**Keywords:** brain-machine interfaces, motor control decoder, spiking neural networks, neuromorphic systems, ECoG, personalized neuro-prosthetics

## Abstract

Current technological solutions for Brain-machine Interfaces (BMI) achieve reasonable accuracy, but most systems are large in size, power consuming and not auto-adaptive. This work addresses the question whether current neuromorphic technologies could resolve these problems? The paper proposes a novel neuromorphic framework of a BMI system for prosthetics control via decoding Electro Cortico-Graphic (ECoG) brain signals. It includes a three-dimensional spike timing neural network (3D-SNN) for brain signals features extraction and an on-line trainable recurrent reservoir structure (Echo state network (ESN)) for Motor Control Decoding (MCD). A software system, written in Python using NEST Simulator SNN library is described. It is able to adapt continuously in real time in supervised or unsupervised mode. The proposed approach was tested on several experimental data sets acquired from a tetraplegic person. First simulation results are encouraging, showing also the need for a further improvement via multiple hyper-parameters tuning. Its future implementation on a neuromorphic hardware platform that is smaller in size and significantly less power consuming is discussed too.

## 1. Introduction

Progress in neuroscience, biomedical engineering, and artificial intelligence led to significant advancements in Brain-Computer Interface (BCI) technology and its applications in various domains. Contemporary signal acquisition and processing techniques significantly improved the accuracy and reliability of BCIs [1,2]. In particular, artificial intelligence (AI) offered novel neural signal decoding techniques, including adaptive algorithms and deep learning models with improved classification accuracy. Recent medical applications of BCI offer revolutionary benefits to patients with neurological conditions such as stroke, spinal cord injuries, and neurodegenerative disorders [3,4] leading to severe communication disabilities, allowing direct interaction with devices and prostheses.

The research and practical applications of BMIs for controlling human prosthetics have been developing fast in the last decade [5,6,7,8,9]. A pioneering work has been developed in [10,11], where brain implants are inserted in the skull to measure epidural ElectroCorticoGrams (ECoG) signals of the motor and sensory cortices in order to extract movement intentions. These signals are decoded in a motor control decoder (MCD) using machine learning and statistical signal processing techniques. The output of the decoder is sent as commands to control exoskeleton or implantable spinal cord stimulator. For a better calibration, in [12] an adaptive tensor-based recursive exponentially weighted Markov-switching multi-linear model (REW-MSLM) decoder is proposed. REW-MSLM uses a mixture of expert (ME) architecture, mixing or switching independent decoders (experts) according to the probability estimated by a ‘gating’ model. The performance of the REW-MSLM decoder allows the patients to use the neuroprosthesis for several weeks without recalibration. In this case, an unsupervised recalibration/adaptation would be a great added value for home use.

However, there are still many challenges and unsolved problems that prevent the commercialization of advanced BCI systems, such as accurately interpreting brain signals, the need for individual calibration, and ensuring long-term reliable use. The NEMO-BMI project (https://nemo-bmi.net, accessed on 9 March 2025) aims at the development of on the one hand auto-adaptive spinal stimulation and brain signal decoders and on the other hand of miniaturized and low power hardware for brain signal decoding. MCDs and their hardware realizations using both classical machine learning and statistical signal processing methods along with neuromorphic methods are part of the project.

With the development of neuromorphic devices, that are small in size, consume significantly less power and are adaptable to new data, such as SpiNNaker [13] and Loihi [14], to mention only two of them, the challenge now is to develop neuromorphic computational models for BMI and to implement them on these devices. Spike timing neural networks (SNN) are the most used and implemented in the neuromorphic devices neural information processing techniques, where brain-inspired principles of machine learning, such as spike information representation and spike timing dependent plasticity (STDP) are used along with many other methods [15].

There are few works on ECoG signals decoding due to limited availability of experimental data. As far as we know the SNN and ESN were applied separately to such task [16,17]. However, no similar works combining both neural network architectures. The proposed combination of SNN and ESN was initially tested on a much simpler task for EEG signals decoding and demonstrated promising results [18]. The current work is first application of this neuromorphic structure to experimental data sets from ECoG device obtained from implanted tetraplegic patients.

Our paper aims at the introduction of a neuromorphic framework using SNNs, which allows adaptation of the neuronal activity decoder in a real time during the neuroprosthesis self-directed use. The neuromorphic algorithms may improve BMI performance and can be implemented in a novel chip with low-power consumption compared to conventional BMI algorithm hardware implementations.

This paper adds numerous discussions on the challenges related to the development of adaptive BMI [19,20,21,22,23,24], by proposing a framework of a neuromorphic MCD along with the discussions on the applicability of various design methods. The addressed issues and the proposed framework are illustrated on a subset of data from case study [10,12]. These include ECoG data of a single person for idle vs several movement states either in serious game or in real 3D space. The used in this paper ECoG data is collected using 64 electrodes as described in [10,12]. Although the hand and leg movements are triggered by brain activity in numerous spatial clusters of neurons over time, here we consider only ECoG signals from the motor cortex, which makes the task of predicting the desired by a patient’s movement challenging.

The pioneering designed and experimentation of MCD in CEA [10,12] has demonstrated a good accuracy of detecting imaginary movements of legs and hands. In this paper, we are aiming at achieving similar results, but with the use of neuromorphic algorithms that would allow conceptually for a small in size and drastically lower power consuming BMI in the future.

## 2. Materials and Methods

### 2.1. Neuromorphic Framework for Adaptive MCD

One class of computational SNN architectures are the brain-inspired SNN, where a SNN is spatially structured using a three-dimensional (3D) brain template, and connectivity trained on spatio-temporal brain data, such as EEG or ECoG using STDP or other brain-inspired learning methods [15]. In the NeuCube SNN architecture [25], the spatio-temporal brain patterns learned in such a 3D SNN are classified or used for regression in an output SNN module.

A new computational framework is proposed here, where a 3D-SNN is structured according to an individual brain template.

The patterns learned in the SNN from the ECoG time series data are neurons’ spiking frequencies during a given period of time. They are classified by a recurrent fast trainable neural network architecture called Echo State Network (ESN). A preliminary version of this framework was first tested on a benchmark EEG data, producing better classification results than an original system [18]. Here, this framework is further developed for MCD on ECoG signals and tested on the case study data from [10,12]. The proposed framework is shown in Figure 1. It consists of two basic modules:A 3D recurrent SNN architecture (3D-SNN), spatially structured and adaptable to an individual 3D brain template, for feature extraction from recorded ECoG brain signals. The SNN is incrementally trained in an unsupervised mode.A recurrent Echo State Network (ESN) for decoding of the brain trained in a supervised mode on-line via recursive least squares (RLS) or reinforcement learning (RL) algorithms.

The 3D-SNN is structured according to a personalized brain template of a patient. It can consist of full 3D brain volume template or only of the 3D ECoG electrodes positions. The personalized 3D SNN was designed from a 3D scan of a patient’s brain. The individual positions of neurons were set to fit within the 3D brain structure. SNN consists of leaky integrate and fire (LIF) neurons modeled by Equation (Equation 1):(1)$$dV_{m}/dt=-\left(V_{m}-E_{L}\right)/\tau_{m}+\left(I_{syn}+I_{e}\right)/C_{m}$$
where $V_{m}$ is neural cell membrane potential; $E_{L}$—resting membrane potential; $\tau_{m}$—membrane time constant; $C_{m}$—membrane capacity; $I_{e}$—constant input current; $I_{syn}$—total synaptic current coming from connected neurons. A spike is emitted when cell membrane potential $V_{m}$ goes above the threshold value $V_{th}$ and then the membrane potential returns to its resting state $V_{rest}$ for refractory period $t_{ref}$. The hyper-parameters of this structure include individual neurons’ model parameters such as spiking threshold of the membrane potential and refractory time.

Neurons were positioned in the 3D space according to the template. The overall connectivity within 3D-SNN is initially designed based on small world connectivity rule, i.e., the close in 3D space are neurons the stronger (with bigger weight) is their connection and the weight strength decreases exponentially with distance increase. In order to allow for continuous auto-adaptation of this structure Spike timing dependent plasticity (STDP) connections, called synapses, were adopted. The neuronal activity of the 3D-SNN is triggered by filtered ECoG electrodes measurements called further temporal features. In current study these temporal features were extracted from raw ECoG data by Morlet transformation with various fundamental frequencies [20] or by shifting the row signal. They are fed as generating currents of the neurons corresponding to ECoG electrode positions. The 3D-SNN connections adapt continuously to these signals via STDP rule [26] in unsupervised mode. The STDP mimics the way our brain adjusts the strength of connections between pairs of neurons based on the relative timing of pre- and post-synaptic neuron’s reaction (spikes), reflecting the activity-dependent plasticity of nervous systems. Thus, our 3D-SNN model continuously adapts connection weights based on incoming signals reflecting the patient’s ECoG records. A hyper-parameter defining the speed of this adaptation is STDP learning rate λ.

The features extracted form 3D-SNN are spiking frequencies of the selected neurons (in our case those corresponding to the ECoG electrode positions) for a given time interval (59 recording steps in our case study below) denoted as $t_{ECoG}$:(2)$$features\left(t-t_{ECoG}÷t\right)=n_{spikes}⁄t_{ECoG}$$

Next, these features are fed as input in(t) to the ESN, that is a member of a novel and rapidly developing family of reservoir computing approaches [27], whose aim is the creation of a fast trainable recurrent neural network (RNN) architecture, able to approximate nonlinear time series. It incorporates a pool of neurons with sigmoid activation function (usually the hyperbolic tangent) that has randomly generated recurrent connection weights. The reservoir state R(t) for the current time instant *t* depends both on its previous state R(t−1) and the current input in(t) as follows:(3)$$R\left(t\right)=\left(1-a\right)R\left(t-1\right)+a\tanh(W^{in}in\left(t\right)+W^{res}R\left(t-1\right))$$

Here $W^{in}$ is the matrix of input to reservoir connection weights that are randomly generated; $W^{res}$ is the internal reservoir connection weight matrix that is sparse and also randomly generated, with spectral radius below 1; *a* is leaking rate parameter.

The ESN output out(t) is calculated as a linear combination of the concatenation of the input and reservoir states [R(t)in(t)] weighted by the output weight matrix $W^{out}$:(4)$$out\left(t\right)=W^{out}[\left(R\left(t\right)\;in\left(t\right)\right]$$

The only trainable parameters of ESN are the output weights $W^{out}$. Since the output is linear function, the least squares method is applied to train the ESN in a single iteration. For the aims of on-line training of our model the recursive version of least squares (RLS) was applied.

The ESN configuration is defined by the following hyper-parameters: number of neurons in the reservoir pool, sparsity of recurrent connectivity matrix $W^{res}$ (this means number of non-zero weighted connections) and leaking rate parameter a that defines how much previous reservoir state R(t−1) contributes to its new state R(t) according to Equation (Equation 3).

The ESN of our MCD has different groups of outputs and corresponding to them parameters $W^{out}$ that are presented below in the case of the different experimental data sets.

State (idle, hand or leg movement) ($out\left(t\right)=state,\;W^{stateout}$): a vector with dimension corresponding to the number of possible states that was decoded via argmax from the predicted model output.Trajectory (left and right hand positions in 3D space) ($out\left(t\right)=trajectory,\;W^{trajectory}$): a vector of *x*, *y* and *z* coordinates for each hand.Satisfaction ($out\left(t\right)=satisfaction,\;W^{satisfactionout}$): two-dimensional vector that was decoded via argmax from the predicted model output.

A satisfaction corresponds to when an action performed by the BCI’s effector matches the user’s intention. A dissatisfaction is when it does not match. In [28,29], the authors used the “satisfaction” to derive motor labels in order to auto-adaptively update their motor control decoder. In presented here case, the satisfaction was used as reinforcement signal in order to correct the output of MCD in case of dissatisfaction.

The adaptation of the proposed MCD is achieved in two ways, as follows:Using STDP rule to continuously adapt the 3D-SNN connectivity in an unsupervised way, based on history of the input signal as well as the state of spiking neurons. This can be used for auto-adaptation of the model, when new data from the subject are continuously entered for unsupervised learning. This will ensure that the model will continuously adapt (auto-adapt) with the subject’s movement improvement in time.Using reinforcement learning rule (RL) of the output parameters of the ESN responsible for motor action based on satisfaction prediction.

### 2.2. Software Implementation

The training algorithm is shown as Algorithm 1. In training mode the only adjustable parameters of the model are the output connection weights $W^{out}$ of the ESN module. They are tuned incrementally with every new input/output training data pair via recursive least squares (RLS) method. The 3D-SNN connection weights change according to STDP rule continuously.
**Algorithm 1** Pseudocode of on-line training algorithm**Initialization**
Initialize ESN module parameters
Compose 3D-SNN module using neurons and ECoG positions
  1:**while** 
newdata 
**do**  2:   Readfromfiles←ECoGsignalsanddesiredstate  3:   generatingcurrents←filteredECoGsignals  4:   **for** $t_{ECoG}$ **do**  5:       Simulate3D−SNN  6:   **end for**  7:   features←3D−SNN  8:   ESN←features  9:   predictedstate←ESNoutput10:   $train\;W^{out}$11:**end while**12:Keepmodelparameters


In case of dissatisfaction predicted (or expressed by the patient if possible) reinforcement learning rule [30] changes the output parameters of MCD ($W^{out}$) using “satisfaction” label as reinforcement signal as follows:(5)$$\begin{array}{c} W^{out}=W^{out}-\alpha \left(1-satisfaction\right)E\\ E=\delta E+(1-\delta )[R\;in]out \end{array}$$
where α and δ are learning rate and eligibility trace *E* decay rate respectively; *R*, in and out are current state, and output of the ESN module.

A block scheme of the developed neuromorphic algorithm for the MCD is shown in Figure 2. Both modules are written in Python, version 3.8.9. The 3D-SNN is based on NESTsimulator library, version 3.3 [31] while the rest of code exploits numpy, SciPy and other Python libraries for mathematical calculations. The software works in pseudo-online mode receiving (in this version, simply reading from edf file) as input the next portion of 59 records from ECoG electrodes and generating as output motor actions as well as satisfaction label for current time step. Results are kept in csv files but could be easily exported in a pseudo-online mode one by one.

Preliminary studies have been conducted on the feasibility of implementing the proposed MCD framework and algorithms on a neuromorphic hardware platform. The estimation of model running time was done on a desktop configuration with following parameters:12th Gen Intel® Core™ i7-12700Installed RAM 32 GBBase speed 2.10 GHzCores 12Logical processors 20

The software runs under WSL in Windows 11. Estimated times for model initialization and during the training and testing phases are as follows:Initialization includes generation/reading of model parameters and setting-up of two recurrent structures (3D-SNN and ESN): 49 s approximately.Single RLS iteration (training on one input/output train data example): 2.5–3.5 sSingle RL iteration (training on one input/output train data example): 0.55–0.65 sSingle step model output calculation: 0.58–0.63 s

Our preliminary tests on neuromorphic hardware foreseen in this project—Intel’s Loihi—showed the following speed of 3D-SNN structure only: simulation for 10,000 ms of 1000 sparsely connected LIF neurons execution time was 55.89 ms (using the built-in Liohi2 profiler function). In comparison, our tests on a single core virtual machine on desktop configuration with Intel Core™ i3-4130 CPU, 3.4 GHz, 8 GB RAM needed approximately 1 s to simulate for 1000 ms a 3D-SNN with 1471 neurons sparsely connected via STDP synapses using NEST simulator library. Thus, Loihi2 would offer approximately 200 (1000/5) times faster execution time and of course much lower power consumption.

At this point, we have a 3D-SNN implementation in Lava-0.3.0 (software recommended from Intel to prepare a code of implementation in Loihi). ESN module can be implemented also on the same neuromorphic hardware. As currently our model has about 1600 neurons totally (3D-SNN + ESN). We expect execution time on neuromorphic hardware for a test step (that uses ECoG recordings from approximately 100 ms) to be orders of magnitude faster than on a desktop computer, e.g., approximately 0.6/200 s.

### 2.3. Experimental Data

The described here MCD experimental model, based on the proposed MCD framework, was trained on three data bases (DBs) collected from two different male patients with traumatic sensorimotor tetraplegia caused by a complete C4–C5 spinal cord injury as described in [10,12].

First patient underwent bilateral, while the second one—unilateral (on left hemisphere) implantation of chronic wireless ECoG WIMAGINE implants [32] composed of 64 planar electrodes with a 2.3 mm active diameter and 4–4.5 mm inter-electrodes distance (electrodes are in contact with the dura mater). The ECoG signals are low and high pass filtered in a bandwidth from 0.5 Hz to 300 Hz using analog bandpass filter and, after digitalization, using a digital low pass FIR filter with a cutoff frequency of 292.8 Hz, for digitalization the sampling frequency is 585.6 Hz, the resolution chosen at 10 bits in the Analog Digital Converter and eventually radio-transmitted in the MICS band (402–405 MHz) to a custom base station connected to a computer [32]. During the experimental sessions, 32 electrodes for each implant were selected in a checkerboard-like pattern because of limited data rates of the Microsemi Zarlink radio link (250 kb/s in 2-FSK mode).

First DB, called the “Runner”, was described in [28,29]. In this experiment, the first test subject was seated in front of a computer screen where a human avatar was represented from a third person perspective. The avatar could either stand still or walk forward at a fixed speed. Figure 3 shows the experimental set-up. The subject controlled the avatar using leg motor imagery. 13 sessions, of on average 11 min of recording, were acquired over 141 days. For blind-test of the algorithms, the database has been split in two parts. The first 9 sessions are provided with ECoG data and labels, while the 4 last contain ECoG data and the satisfaction labels only.

The second DB, called 6DoF, is described in Alexandre Moly et al., 2022 [12]. BCI experimental data was acquired during experimental sessions where the first patient performed 6D control of a virtual avatar with 3 different states: move right arm (3D movement), move left arm (3D movement) and idle. Figure 4 shows the experimental set-up. The hand trajectories were described in 3D space by consecutive positions (3D coordinates). 16 sessions were recorded. For blind-test of the algorithms, the database has been separated in two parts. The first 10 sessions are provided with labels and ECoG data, while the 6 last sessions are used for a blind test.

The third DB called 5DoF was collected during the experiment with the second subject. He was seated in front of a computer screen showing a virtual environment with avatar having four possible movement states in addition to its idle state: grasping by the right hand, grasping by the left hand, flexion of the right elbow and flexion of the left elbow. The subject performed direct motor imagery to activate each of the motor states. Two sessions of 32 and 53 min were performed. For blind-test of the algorithms, some labels were removed from the database. Both sessions are provided with ECoG data, labels (when not removed), satisfaction labels, and the three flags of updates (for reinforcement and supervised training of motor decoder as well for updating of satisfaction decoder).

## 3. Results

This section shows blind tests accuracy assessment of experimental MCD models trained on the three DBs described above. Here we also compare the two approaches for extracting ECoG temporal features, namely:Approach1: Shifting up amplitudes of the ECoG signals.Approach2: Square Morlet transformation [33] of ECoG signals using 15 fundamental frequencies (from 10 to 150 Hz with a step of 10 Hz).

The model hyper-parameter values are given in Table 1. Some of them were chosen by trail and error, others will be subject of further tuning.

Figure 5, Figure 6 and Figure 7 present accuracy of trained MCD models on Runner DB, 6DoF DB and 5DoF DB respectively.

The evaluation of features extraction approaches on Figure 5 demonstrates that Morlet wavelet transform (Approach2) improved MCD prediction accuracy up to about 65% for some fundamental frequencies (e.g., 30 Hz and 40 Hz) in comparison with Approach1 for which results are slightly above chanse level of 50%. T-test also demonstrated that the accuracy difference between models trained via two compared approaches is statistically significant (*p*-value strongly below 0.05). “Voting” models decide using models trained with features extracted by several or all Morlet fundamental frequencies. The “voting” means that final decision was taken based on the decision of the majority of models selected to vote. In case of “voting all” all models using all fifteen fundamental Morlet frequencies were accounted for while in case “voting chosen” six out of fifteen models (for fundamental frequencies of 10, 30, 50, 70, 100 and 110 Hz) were selected. The decision threshold (minimal number of models that has to agree on decision) was selected based on final decision accuracy. In case of “voting all” it was determined to be 14 out of 15 models while in case of “voting chosen” it was 3 out of 6 models. We observe that the best results were obtained by voting among the models using selected best fundamental frequencies. T-test confirmed statistically significant difference between model trained on raw signal and “voting chosen” model (*p*-value 0.00009) while the “voting all” model accuracy does not differ significantly from raw signal model (*p*-value 0.289).

Table 2 summarizes mean and variance of balanced accuracies as well as of Fscores on both classes (idle and walk) for runner DB models. According to mean values of the balanced accuracy and $F_{scores}$ on both classes the best models are those using Morlet transformation at 20 and 30 Hz fundamental frequency. Voting among all models shows increase of $F_{score}$ on idle class at the expense of significant decrease of $F_{score}$ on class walk while voting of several chosen models demonstrates the best results on mean balanced accuracy as well as on $F_{scores}$ on both classes.

$F_{score}$ on idle class is much higher than on class walk, which can be explained by imbalanced number of examples from idle vs walk classes in this DB. The combination of features extracted via multiple fundamental frequencies could probably result in better classification accuracy at the expense of increased model dimension. Further investigation of 15 models and their features would allow us to optimize the model.

As it can be seen on Figure 5 and in Table 2, the test accuracy of a MCD model based on Morlet transformation of ECoG signals depends on the fundamental frequency used. As demonstrated there, a comparative analysis is needed for the selection of the best input features from raw ECoG signals for a particular subject, before a 3D SNN model is trained on these features for this subject. Hence a preliminary analysis of ECoG data collected from a new subject and optimal feature selection is a crucial step towards the design of an efficient neuromorphic MCD.

Results about 6DoF data base shown on Figure 6 were obtained using Morlet transform (Approach2 for features extraction) at 10 Hz only. Further tests with other fundamental frequencies of Morlet wavelet would result in better accuracy as in previous DB tests. The idle state $F_{score}$ may be improved after balancing the data. Cosine similarity is used to measure both hands’ trajectory correctness. It shows the need for improvement that can be achieved through further optimization of hyperparameters as discussed below.

Figure 7 shows testing accuracy of an exemplar MCD on 5DoF DB using Approach2 for features extraction. These results demonstrated accuracy that is above chance level, however it also shows the need for further improvement. Since in this DB ECoG signals were obtained only from a single-side (e.g., left) implant, we expect that a higher accuracy could be achieved with the use of two implants, located on the left and the right parts of the motor cortex Besides, these results were obtained using Morlet transform with 30 Hz fundamental frequency, so further investigation of other frequencies as well as a voting model might increase the testing accuracy. Another direction for improvement of results could be changing the training approaches, since we apply both RLS and RL algorithms on the ESN in dependence on flags in DB.

## 4. Discussion

Comparison with MCD accuracy on the same data sets reported in [10,12,28,29] shows the need of improvement of our model. Neverthless, our approach has potential to increase MCD prediction accuracy subject to refinement of its multiple hyper-parameters.

The assessment of the neuromorphic model accuracy presented here was done on a given set of hyper-parameter values. Some of them were determined based on our preliminary grid search on much smaller data set. The benchmark data set (EEG records for three different movements, 1 s per record, 60 examples, 20 per class) we had to test totally 243,000 combinations that took about 9 days on a supercomputer platform [18].

With much bigger data from NEMO-BMI project such tuning would need enormous time so the hyper-parameters in the tests on three DBs reported here were chosen based on preliminary results from [18]. Further fine tuning of hyper-parameters using different methods (e.g., evolutionary algorithms [15]) would result in higher model accuracy.

## 5. Conclusions

The paper presents a novel approach for the design of a personalized neuromorphic motor control decoder (MCD). The proposed MCD framework consists of modules that can be performed with the use of different methods and their parameters depending on the task and the available personal data.

The practical use of the framework requires analysis and understanding of the personalised data, selection of features, training a 3D SNN and an output classifier/regressor, along with optimization of the parameters of a model for a better performance for the individual in question. Using different methods for classification and prediction and rigorous parameter optimization could have an impact on the overall accuracy of the created personalized MCD.

We have to admit also the big challenge to recruit a larger database since ECoG implanted patients are few in the world. Since these are mainly people with disabilities, carrying out of experiments to collect data with them is subject of strict medical control.

The paper presents a software written in NEST and Python and discusses its possible implementation on a neuromorphic platform. As the proposed MCD framework is open for adding new methods to accomplish different functionality, further research is needed to establish the neuromorphic approach for efficient design of future MCD, that are minimal in size and highly power efficient.

## Figures and Tables

**Figure 1 biomimetics-10-00183-f001:**
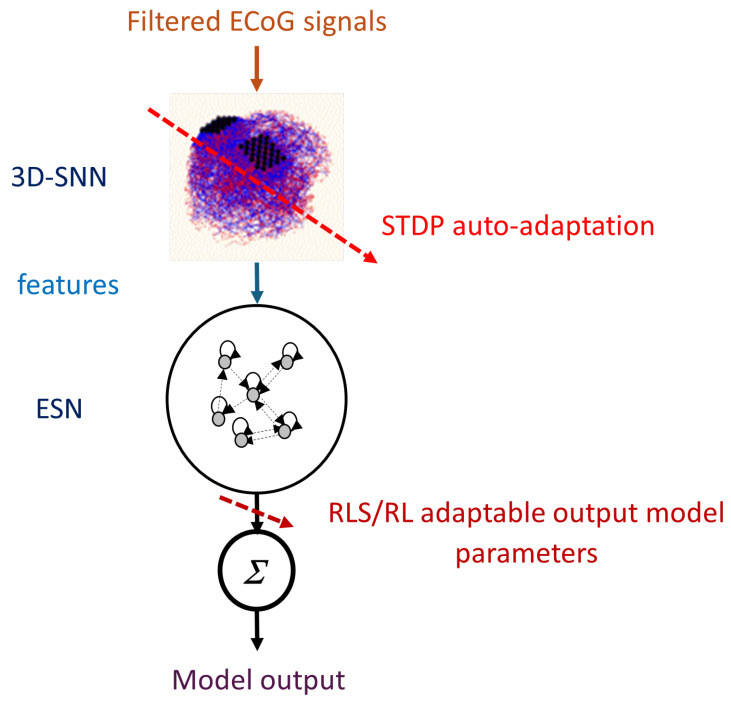
Functional diagram of the proposed MCD.

**Figure 2 biomimetics-10-00183-f002:**
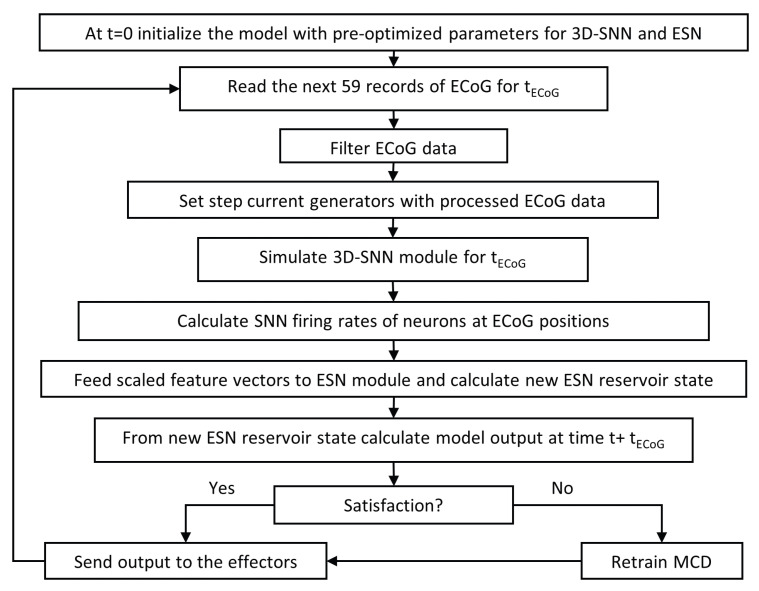
The overall algorithm of the MCD framework.

**Figure 3 biomimetics-10-00183-f003:**
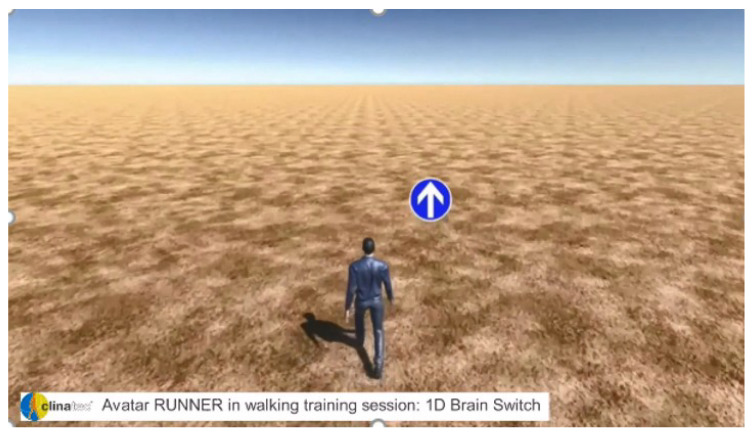
Experimental set-up for Runner DB acquisition. The patient seats in front of a screen and tries to control via his ECoG signals an avatar to walk or to stay idle.

**Figure 4 biomimetics-10-00183-f004:**
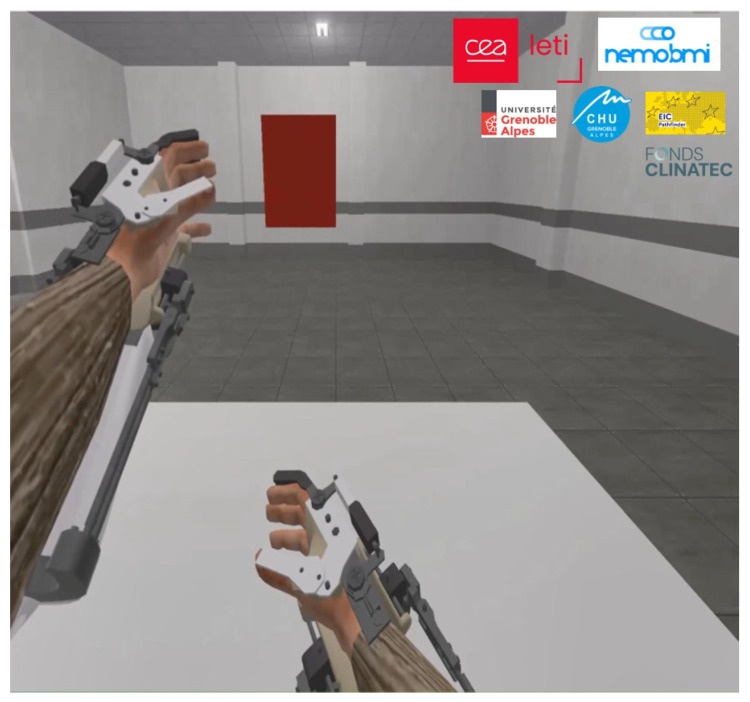
Experimental set-up for 6DoF DB acquisition. The patient is asked to control the virtual avatar with 6 degrees of freedom to move his left or right hand or to stay idle. Both hands trajectories are described by sequence of target positions defined by their coordinates in 3D space.

**Figure 5 biomimetics-10-00183-f005:**
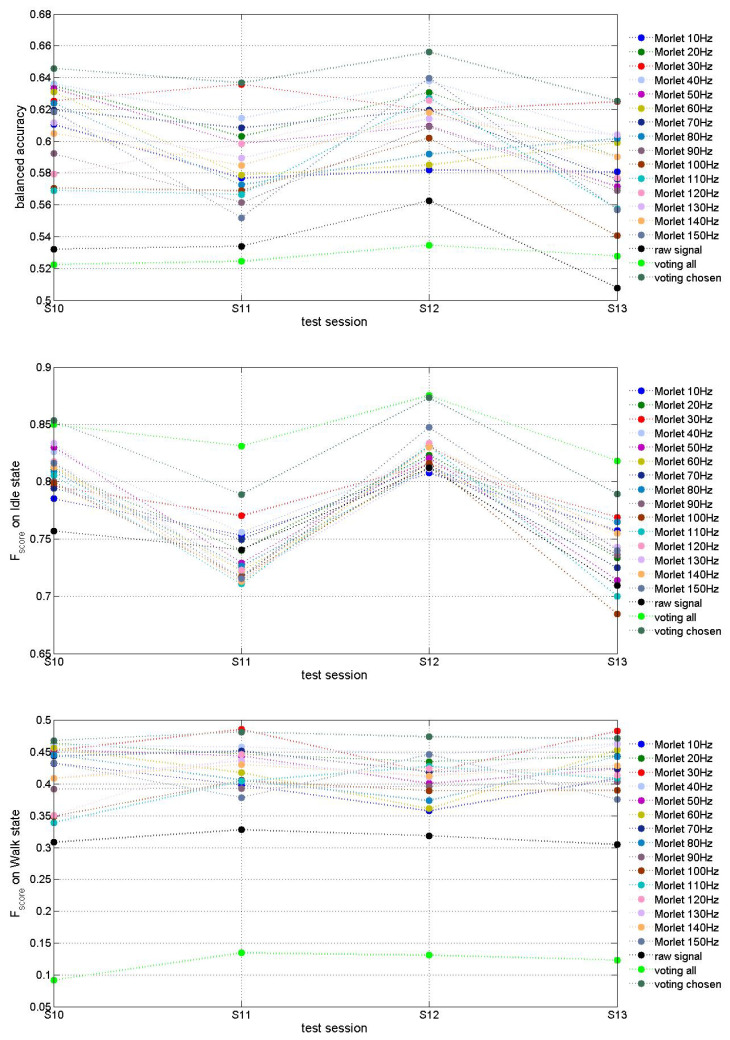
Testing accuracy of exemplar MCD on Runner DB. The figure on the top presents the balanced accuracy of the model predictions while the middle and bottom figures present $F_{score}$ on predictions of the two classes (idle and walk) respectively.

**Figure 6 biomimetics-10-00183-f006:**
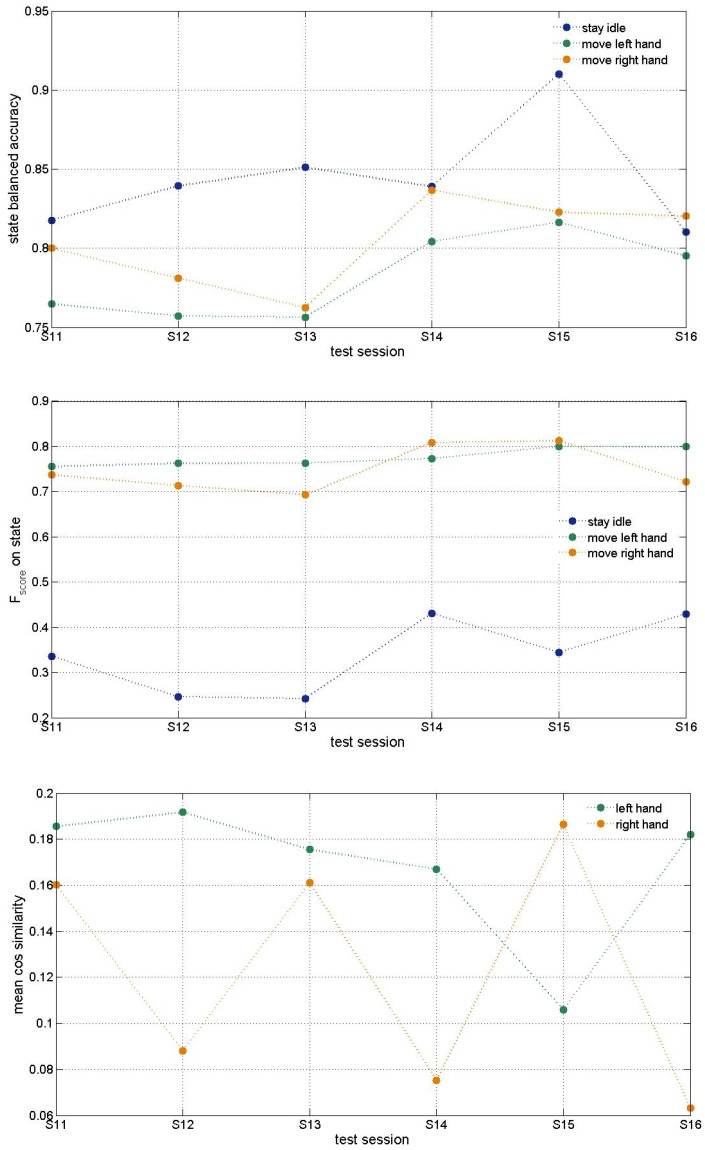
Testing accuracy of exemplar MCD on 6DoF DB. The top figure presents accuracy of prediction of the three classes of the state desired (idle, move left hand, move right hand). Middle plot shows $F_{score}$ on three state classes. The bottom plot shows mean cosine similarity per test session of the predicted vs desired hand trajectory.

**Figure 7 biomimetics-10-00183-f007:**
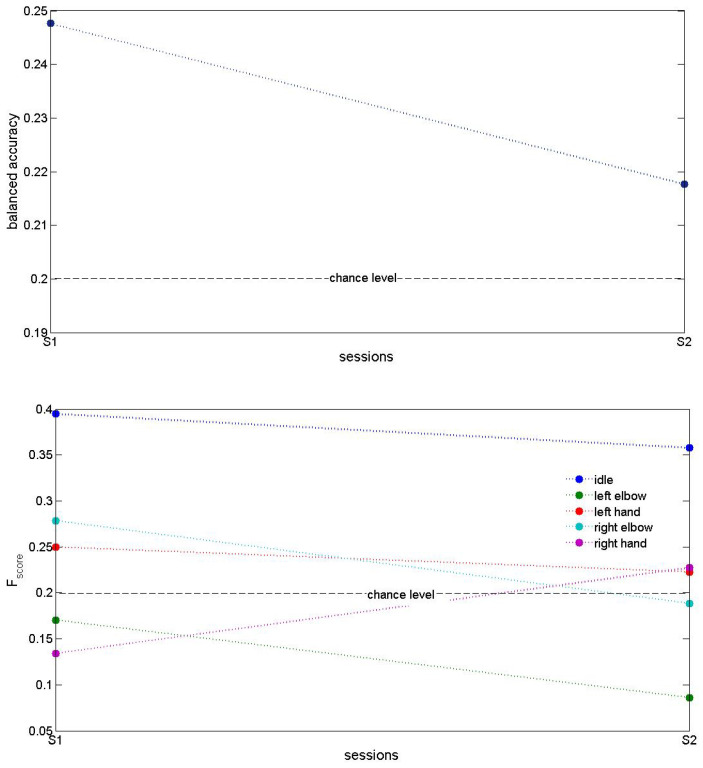
Testing accuracy of an exemplar MCD on 5DoF DB. Top plot shows balanced accuracy of five states (idle, move left elbow, left hand, right elbow or right hand) predictions for the two test sessions. The bottom plot shows $F_{score}$ on five classes.

**Table 1 biomimetics-10-00183-t001:** Exemplar model hyper-parameters.

Hyperparameter	Value
3D-SNN	
membrane potential threshold $V_{th}$	−65.0 mV
refractory time $t_{ref}$	0 ms
STDP learning rate λ	0.001
ESN	
leaking rate *a*	0.5
number of reservoir neurons	15,000
sparsity of reservoir connectivity matrix $W^{res}$	0.5

**Table 2 biomimetics-10-00183-t002:** Mean values of the balanced accuracy and $F_{scores}$ on classes idle and walk for testing data sessions of all models from Figure 5. Best results are marked in bold.

Model	Mean Balanced Accuracy	STD of Balanced Accuracy	Mean $F_{\mathrm{score}}$ on Idle	STD of $F_{\mathrm{score}}$ on Idle	Mean $F_{\mathrm{score}}$ on Walk	STD of $F_{\mathrm{score}}$ on Walk
Raw signal	0.534	0.022	0.755	0.043	0.315	0.011
Morlet 10 Hz	0.587	0.016	0.776	0.026	0.399	0.031
Morlet 20 Hz	0.615	0.022	0.777	0.047	0.447	0.012
Morlet 30 Hz	**0.626**	0.007	**0.787**	0.022	**0.460**	0.032
Morlet 40 Hz	**0.623**	0.017	**0.785**	0.045	**0.457**	0.008
Morlet 50 Hz	0.603	0.026	0.773	0.060	0.431	0.024
Morlet 60 Hz	0.599	0.023	0.772	0.047	0.422	0.044
Morlet 70 Hz	0.606	0.021	0.771	0.041	0.435	0.015
Morlet 80 Hz	0.598	0.021	0.778	0.040	0.417	0.034
Morlet 90 Hz	0.583	0.022	0.771	0.053	0.396	0.005
Morlet 100 Hz	0.571	0.025	0.754	0.063	0.383	0.024
Morlet 110 Hz	0.580	0.032	0.762	0.066	0.395	0.038
Morlet 120 Hz	0.595	0.022	0.779	0.055	0.409	0.041
Morlet 130 Hz	0.605	0.011	0.776	0.056	0.430	0.025
Morlet 140 Hz	0.600	0.015	0.778	0.054	0.420	0.011
Morlet 150 Hz	0.592	0.044	0.780	0.062	0.409	0.036
Voting all	0.527	0.005	0.844	0.025	0.120	0.019
Voting chosen	**0.641**	0.013	**0.826**	0.044	**0.474**	0.006

## Data Availability

No new data were created or analyzed in this study. This study presents only the MCD model structure and accuracy assesment by simulations.

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
