# Peer review of "Decoding Brain Signals in a Neuromorphic Framework for a Personalized Adaptive Control of Human Prosthetics"

_biomimetics, 2025, doi:10.3390/biomimetics10030183_

Round 1

Reviewer 1 Report

Comments and Suggestions for Authors

The authors have proposed a framework to decode brain signals for prosthetics control purposes. Although the work needs further improvements, it is interesting and the manuscript is solid. I only have a few minor comments for the authors to help improving the quality of the manuscript.

  • Figure panels need labels. Also, they need to be explained a bit in the caption.
  • Some paragraphs are too short. For example, the introduction starts with a single-sentence paragraph. I suggest revising the text a little bit.
  • I do not suggest providing references in the Conclusions section.
  • Some references come with doi while some others don’t. I suggest making them consistent.

Author Response

Comment 1: Figure panels need labels. Also, they need to be explained a bit in the caption.

Answer 1: We have labelled each panel and expanded figure captions accordingly

Comment 2: Some paragraphs are too short. For example, the introduction starts with a single-sentence paragraph. I suggest revising the text a little bit.

Answer 2: We have merged smaller paragraphs and have revised the text.

Comment 3: I do not suggest providing references in the Conclusions section.

Answer 3: We have removed references in the conclusion section.

Comment 4: Some references come with doi while some others don’t. I suggest making them consistent.

Answer 4: References are unified.

Reviewer 2 Report

Comments and Suggestions for Authors

In this study, the authors proposed a neuromorphic framework for real-time prosthetics control through the decoding of Electro Cortico-Graphic (ECoG) signals. The framework utilized a three-dimensional spike timing neural network (3D-SNN) for feature extraction from brain signals and an online trainable recurrent reservoir structure (Echo state network, ESN) for motor control decoding (MCD). The article is clear, concise, and aligns well with the journal's scope. However, there are some details that should be checked:

  1. Suggest the authors review the data usage description carefully. For example, according to the text, in the second DB, 16 sessions were recorded, but only the first 10 and the last 5 sessions were used. However, Figure 5 suggested that session 11-16 were used for testing.
  2. Suggest the authors provide more details on the methodology part for a better understanding. For example, how the models were selected in ‘voting chosen’ in Figure 5, and the basic setup for the pseudo-online mode.
  3. Suggest the authors include more quantitative results in the main text to strengthen the analysis and enhance the overall cogency and rigor of the article.

Author Response

  1. Suggest the authors review the data usage description carefully. For example, according to the text, in the second DB, 16 sessions were recorded, but only the first 10 and the last 5 sessions were used. However, Figure 5 suggested that session 11-16 were used for testing.

Thank you for noticing. The typo mistake was corrected. Test sessions are indeed 6, not 5.

  1. Suggest the authors provide more details on the methodology part for a better understanding. For example, how the models were selected in ‘voting chosen’ in Figure 5, and the basic setup for the pseudo-online mode.

We added the following explanation text:

The “voting” means that final decision was taken based on the decision of the majority of models selected to vote. In case of “voting all” all models using all fifteen fundamental Morlet frequencies were accounted for, while in case of “voting chosen”, six out of fifteen models (for fundamental frequencies of 10, 30, 50, 70, 100 and 110 Hz) were selected. The decision threshold (minimal number of models that has to agree on decision) was selected based on final decision accuracy. In case of “voting all” it was determined to be 14 out of 15 models while in case of “voting chosen” it was 3 out of 6 models.

  1. Suggest the authors include more quantitative results in the main text to strengthen the analysis and enhance the overall cogency and rigor of the article.

The additions into the old text (in blue) are marked in red as follows:

The evaluation of features extraction approaches on Figure 5 demonstrates that Morlet wavelet transform (Approach 2) improved MCD prediction accuracy up to about 65% for some fundamental frequencies (e.g. 30Hz and 40Hz) in comparison with Approach 1 for which results are slightly above chance level of 50%. T-test also demonstrated that the accuracy difference between models trained via two compared approaches is statistically significant (p-value strongly below 0.05). "Voting" models decide using models trained with features extracted by several or all Morlet fundamental frequencies. The “voting” means that final decision was taken based on the decision of the majority of models selected to vote. In case of “voting all” all models using all fifteen fundamental Morlet frequencies were accounted for while in case “voting chosen” six out of fifteen models (for fundamental frequencies of 10, 30, 50, 70, 100 and 110 Hz) were selected. The decision threshold (minimal number of models that has to agree on decision) was selected based on final decision accuracy. In case of “voting all” it was determined to be 14 out of 15 models while in case of “voting chosen” it was 3 out of 6 models. We observe that the best results were obtained by voting among the models using selected best fundamental frequencies. T-test confirmed statistically significant difference between model trained on raw signal and “voting chosen” model (p-value 0.00009) while the “voting all” model accuracy does not differ significantly from raw signal model (p-value 0.289).

Table 2 summarizes mean and variance of balanced accuracies as well as of Fscores on both classes (idle and walk) for runner DB models. According to mean values of the balanced accuracy and Fscores on both classes the best models are those using Morlet transformation at 20 and 30 Hz fundamental frequency. Voting among all models shows increase of Fscore on idle class at the expense of significant decrease of Fscore on class walk while voting of several chosen models demonstrates the best results on mean balanced accuracy as well as on Fscores on both classes.

Table 2. Mean values of the balanced accuracy and Fscores on classes idle and walk for testing data sessions of all models from Figure 5.

Model

Mean balanced accuracy

Std of balanced accuracy

Mean fscore on idle

Std of fscore for idle

Mean fscore on walk

Std of fscore for walk

Raw signal

0.534

0.022

0.755

0.043

0.315

0.011

Morlet 10Hz

0.587

0.016

0.776

0.026

0.399

0.031

Morlet 20Hz

0.615

0.022

0.777

0.047

0.447

0.012

Morlet 30Hz

0.626

0.007

0.787

0.022

0.460

0.032

Morlet 40Hz

0.623

0.017

0.785

0.045

0.457

0.008

Morlet 50Hz

0.603

0.026

0.773

0.060

0.431

0.024

Morlet 60Hz

0.599

0.023

0.772

0.047

0.422

0.044

Morlet 70Hz

0.606

0.021

0.771

0.041

0.435

0.015

Morlet 80Hz

0.598

0.021

0.778

0.040

0.417

0.034

Morlet 90Hz

0.583

0.022

0.771

0.053

0.396

0.005

Morlet 100Hz

0.571

0.025

0.754

0.063

0.383

0.024

Morlet 110Hz

0.580

0.032

0.762

0.066

0.395

0.038

Morlet 120Hz

0.595

0.022

0.779

0.055

0.409

0.041

Morlet 130Hz

0.605

0.011

0.776

0.056

0.430

0.025

Morlet 140Hz

0.600

0.015

0.778

0.054

0.420

0.011

Morlet 150Hz

0.592

0.044

0.780

0.062

0.409

0.036

Voting all

0.527

0.005

0.844

0.025

0.120

0.019

Voting chosen

0.641

0.013

0.826

0.044

0.474

0.006

As it can be seen on Fig. 5 and Table 2, the test accuracy of a MCD model based on Morlet transformation of ECoG signals depends on the fundamental frequency used. As demonstrated there, a comparative analysis is needed for the selection of the best input features from raw ECoG signals for a particular subject, before a 3D SNN model is trained on these features for this subject.

Reviewer 3 Report

Comments and Suggestions for Authors

The authors tackle an interesting challenge in brain-machine interfaces (BMI) to improve performance and improve BMI performance and can be implemented in a novel chip with low-power consumption. The proposed combination of a 3D Spiking Neural Network (SNN) and an Echo State Network (ESN) is a promising approach but there are areas of further development and clarification which could strengthen this paper. 

This reviewer recommends that the authors add in more details on the novelty of this technique especially when applying the SNN and ESN architectures in similar domains. More details on the personalization of the SNN parameters, ESN configuration, training,  would solidify the methdology. 

More details are needed on terminologies which seem unfamiliar - e.g., "auto-adaptive," (line 46) "brain template" (line 82).

While the bulk of the paper is focused on the development of the algorithm, the N=2 user study to validate and personalize raises concerns about the true "personalized" aspect of the work. That being said, this author recognizes that the challenge of recruiting a larger database of users with SCIs and ECOG implanted is challenging and would be helpful to highlight such a fact in the paper itself. 

In the results section, this reviewer recommends adding in statistical significance testing to support claims of performance differences between the different approaches. The graphs in Figure 5 could use more description to improve readability. More details on impact of key hyper-parameters on performance would be valuable in demonstrating the framework's robustness and potential.

Overall, this paper presents an interesting perspective, but could use additional discussion of the points discussed above to improve the quality of the paper. 

Author Response

This reviewer recommends that the authors add in more details on the novelty of this technique especially when applying the SNN and ESN architectures in similar domains.

The following comment and two new references were added to introduction:

There are few works on ECoG signals decoding due to limited availability of experimental data. As far as we know the SNN and ESN were applied separately to such task [16,17]. However, no similar works combining both neural network architectures. The proposed combination of SNN and ESN was initially tested on a much simpler task for EEG signals decoding and demonstrated promising results [25]. The current work is first application of this neuromorphic structure to experimental data sets from ECoG device obtained from implanted tetraplegic patients.

  1. Costa, F., Schaft, E.V., Huiskamp, G. et al.Robust compression and detection of epileptiform patterns in ECoG using a real-time spiking neural network hardware framework. Nat Commun 15, 3255 (2024).
  2. Kim, H.-H., Jeong, J., An electrocorticographic decoder for arm movement for brain–machine interface using an echo state network and Gaussian readout, Applied Soft Computing, vol. 117, 2022, 108393

More details on the personalization of the SNN parameters, ESN configuration, training,  would solidify the methdology. 

The following text (in red) was added to the original text (in blue):

The 3D-SNN is structured according to a personalized brain template of a patient. It can consist of full 3D brain volume template or only of the 3D ECoG electrodes positions. The personalized 3D SNN was designed from a 3D scan of a patient’s brain. The individual positions of neurons were set to fit within the 3D brain structure.

…to its resting state Vrest for refractory period tref. The hyper-parameters of this structure include individual neurons’ model parameters such as spiking threshold of the membrane potential and refractory time.

The overall connectivity within 3D-SNN is initially designed based on small world connectivity, i.e. the closer in 3D space neurons are, the stronger (with bigger weight) is their connection and the weight strength decreases exponentially with the distance between the neurons in the 3D space. In order to allow for continuous auto-adaptation of this structure Spike-Timing Dependent Plasticity (STDP) connections, called here synapses, were adopted.

The 3D-SNN connections adapt continuously to these signals via STDP rule [27] in an unsupervised mode. The STDP mimics the way our brain adjusts the strength of connections between pairs of neurons based on the relative timing of pre- and post-synaptic neuron's reaction (spikes), reflecting the activity-dependent plasticity of nervous systems. Thus, our 3D-SNN model continuously adapts connection weights based on incoming signals reflecting the patient’s ECoG records. A hyper-parameter defining the speed of this adaptation is an STDP learning rate lambda.

The only trainable parameters of ESN are the output weights Wout. Since the output is linear function, the least squares method is applied to train the ESN in a single iteration. For the aims of on-line training of our model the recursive version of least squares (RLS) was applied.

The ESN configuration is defined by the following hyper-parameters: number of neurons in the reservoir pool, sparsity of recurrent connectivity matrix Wres (this means number of non-zero weighted connections) and a leaking rate parameter that defines how much previous reservoir state R(t-1) contributes to its new state R(t) according to eq. (3).

More details are needed on terminologies which seem unfamiliar - e.g., "auto-adaptive," (line 46) "brain template" (line 82).

These issues were explained above.

While the bulk of the paper is focused on the development of the algorithm, the N=2 user study to validate and personalize raises concerns about the true "personalized" aspect of the work. That being said, this author recognizes that the challenge of recruiting a larger database of users with SCIs and ECOG implanted is challenging and would be helpful to highlight such a fact in the paper itself. 

In order to respond to this issue, we’ve added the following sentence in the concluding remarks:

We have to admit also the big challenge to recruit a larger database since ECoG implanted patients are few in the world. Since these are mainly people with disabilities, carrying out of experiments to collect data with them is subject of strict medical control.

In the results section, this reviewer recommends adding in statistical significance testing to support claims of performance differences between the different approaches. The graphs in Figure 5 could use more description to improve readability. More details on impact of key hyper-parameters on performance would be valuable in demonstrating the framework's robustness and potential.

The significance tests were reported above:

T-test also demonstrated that the accuracy difference between models trained via two compared approaches is statistically significant (p-value strongly below 0.05).

 T-test confirmed statistically significant difference between model trained on raw signal and “voting chosen” model (p-value 0.00009) while the “voting all” model accuracy does not differ significantly from raw signal model (p-value 0.289).